# Adiabatic and irreversible classical discrete time crystals

Adrian Ernst, Anna M. E. B. Rossi and Thomas M. Fischer⋆

Experimental Physics X, Universität Bayreuth, 95440 Germany

⋆ thomas.fischer@uni-bayreuth.de

## Abstract

We simulate the dynamics of paramagnetic colloidal particles that are placed above a magnetic hexagonal pattern and exposed to an external field periodically changing its direction along a control loop. The conformation of three colloidal particles above one unit cell adiabatically responds with half the frequency of the external field creating a time crystal at arbitrary low frequency. The adiabatic time crystal occurs because of the non-trivial topology of the stationary manifold. When coupling colloidal particles in different unit cells, many body effects cause the formation of topologically isolated time crystals and dynamical phase transitions between different adiabatic reversible and non-adiabatic irreversible space-time-crystallographic arrangements.

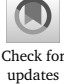

# 1   Introduction

## 1.1   Time Crystals

Time crystals [1,2] are non-equilibrium [3] periodically driven quantum [4,5] or classical [6,7] systems with subharmonic response [8,9]. Discrete time crystals [10–14] are systems coupled to a power supply with a driving frequency that is a higher harmonic of the intrinsic frequency of the isolated few body system. The eldest scientific report of such subharmonic response are the parametric oscillations observed by Michael Faraday in the crispations of a singing wine glass [15]. Such parametric resonance phenomena have been well described with Mathieu's differential equation. Later on discrete time crystals have been realized with phase modulated atomic de Broglie waves or cold atoms [16–23]. Time crystals are also found in an interacting spin chain of trapped atomic ions [5], in a disordered ensemble of about one million dipolar spin impurities in diamond at room temperature [24], in molecular spin systems [25], in an ordered spatial crystal [8], and in a superfluid quantum gas [26]. Recently the possibility of continuous time crystals has been also reconsidered [27,28]. Here we report on a dissipative classical system [29] that lacks an intrinsic frequency such that the driving can be with arbitrary low frequency and the response is always at half the driving frequency:

$$\omega_{\text{response}} = \omega_{\text{drive}}/2 \, . \tag{1}$$

As we show, the reason for such behavior lies in the non-trivial topology of the stationary manifold. In previous work, we have used such non-trivial topology for mesoscopic magnetic colloidal systems [30–35] and macroscopic magnetic systems [36,37] to transport magnetic particles across a periodic pattern. This shows that the topological discrete time crystals shown in this work are intimately connected to other classical [38–44] and quantum mechanical [45,46] topological transport phenomena.

# 2   Adiabatic response

## 2.1   Action-, control-, and product-space, and the stationary manifold

The generalized coordinates $\mathbf{H}_{\mathcal{C}}$ of the drive vary in control space $\mathcal{C}$, while the generalized coordinates $\mathbf{x}_{\mathcal{A}}$ of the few body system vary in what we call the action space $\mathcal{A}$ (see Fig. 1a) and c). The coordinates $\mathbf{H}_{\mathcal{C}}$ in control space (in the colloidal example of section 3 the direction of an external magnetic field) can be manipulated externally and the coordinates in action space $\mathbf{x}_{\mathcal{A}}$ (in section 3 the positions of the colloidal particles) respond to the external modulation. We impose a periodic variation of the coordinates in control space with its trajectory forming a loop $\mathcal{L}_{\mathcal{C}}$. For nonzero response, the potential energy $U(\mathbf{x})$ must couple the driving coordinates

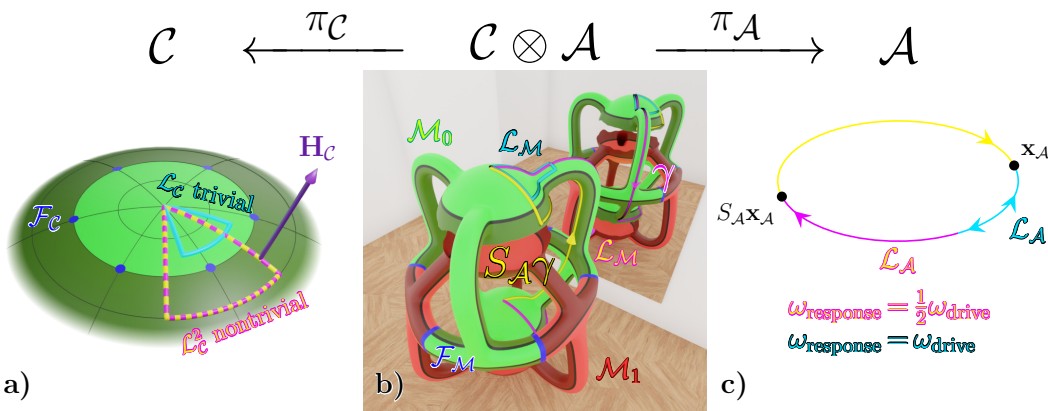

Figure 1: **a)** Control space $\mathcal{C}$, with fence points $\mathcal{F}_{\mathcal{C}}$ (blue) and a trivial (cyan) and non-trivial (yellow/magenta) driving control loop $\mathcal{L}_{\mathcal{C}}$. **b)** The stationary manifold $\mathcal{M}$ in our example is a two dimensional manifold in a three dimensional curved space $\mathcal{C} \otimes \mathcal{A}$. We can visualize the stationary manifold only by deforming it in a way that preserves its topology such that it fits into Euclidian space. Such deformed version is shown in b). Because of the symmetry $S_{\mathcal{A}}$ to each (bright and dark green) region in $\mathcal{C}$ there are two corresponding regions of minima with the same color on the stationary manifold $\mathcal{M}$ in $\mathcal{C} \otimes \mathcal{A}$ and two regions of red color that are maxima. Minima (green) together forming $\mathcal{M}_0$ and maxima (red) forming $\mathcal{M}_1$ of the stationary manifold are separated by fence lines $\mathcal{F}_{\mathcal{M}}$ (blue). Staying on $\mathcal{M}_0$ it is not possible to reach the lower bright green region from the upper without passing through one of the two dark green regions. A trivial loop $\mathcal{L}_{\mathcal{C}}$ in $\mathcal{C}$ causes a trivial loop $\mathcal{L}_{\mathcal{M}}$ (cyan) in $\mathcal{M}$ that does not leave the bright green region. The two symmetric paths $\gamma$ (magenta) and $S_{\mathcal{A}}\gamma$ (yellow) that connect the two different bright green regions of $\mathcal{M}$ via a dark green region are concatenated to form a non-trivial closed loop $\mathcal{L}_{\mathcal{M}}$, that is projected into $\mathcal{L}_{\mathcal{C}}{}^2$ in control space and $\mathcal{L}_{\mathcal{A}}$ in action space. **c)** Action space $\mathcal{A}$ is a non simply connected space and the loop $\mathcal{L}_{\mathcal{A}}$ circulates around a hole of $\mathcal{A}$. Points on opposite sites of action space share the same potential $U(\mathbf{H}_{\mathcal{C}}, \mathbf{x}_{\mathcal{A}}) = U(\mathbf{H}_{\mathcal{C}}, S_{\mathcal{A}}\mathbf{x}_{\mathcal{A}})$. The yellow/pink non-trivial loop $\mathcal{L}_{\mathcal{M}}$ in b) can neither be continuously deformed into a point nor into the cyan trivial loop but can be continuously deformed into one of the fences $\mathcal{F}_{\mathcal{M}}$.

to the coordinates in action space. The pair of both coordinates $\mathbf{x} = (\mathbf{H}_{\mathcal{C}}, \mathbf{x}_{\mathcal{A}})$ therefore varies in the product space $\mathcal{C} \otimes \mathcal{A}$[1]. For a fixed value of the driving coordinate $d\mathbf{H}_{\mathcal{C}}/dt = \mathbf{0}$, i.e. vanishing frequency of the driving modulation $\omega_{\text{drive}} = 0$, the action coordinates remain stationary when residing on the stationary manifold $\mathcal{M} = \{\mathbf{x} \in \mathcal{C} \otimes \mathcal{A} | \nabla_{\mathcal{A}} U(\mathbf{x}) = 0\} \subset \mathcal{C} \otimes \mathcal{A}$. The stationary manifold is the set of coordinates $\mathbf{x}$ for which all forces in action space vanish. The gradient $\nabla_{\mathcal{A}}$ denotes the partial derivative with respect to $\mathbf{x}_{\mathcal{A}}$. Here we show that for quasistatic driving ($\omega_{\text{drive}} \to 0$) it is the topology of the stationary manifold that determines the subharmonicity (equ. 1) of the response in $\mathcal{A}$.

## 2.2 Dissection of the stationary manifold

We call $\mathcal{F}_{\mathcal{M}} = \{\mathbf{x} \in \mathcal{M} | \text{eigenvalue}[\nabla_{\mathcal{A}}\nabla_{\mathcal{A}} U(\mathbf{x})] = 0\} \subset \mathcal{M}$ the fence on $\mathcal{M}$. The fence dissects $\mathcal{M}$ into regions $\mathcal{M}_i, i = 0, 1 \ldots \dim \mathcal{A}$ of different index of the Hessian of the potential, $\nabla_{\mathcal{A}}\nabla_{\mathcal{A}} U(\mathbf{x})$. The index $i$ counts the number of linearly independent directions that are unstable. In the stable stationary manifold, $\mathcal{M}_0$, the index is zero and all forces in the vicinity of

---

[1] The product space $\mathcal{C} \otimes \mathcal{A}$ is defined as the set of coordinates $\{\mathbf{x} = (\mathbf{H}_{\mathcal{C}}, \mathbf{x}_{\mathcal{A}}) | \mathbf{H}_{\mathcal{C}} \in \mathcal{C} \text{ and } \mathbf{x}_{\mathcal{A}} \in \mathcal{A}\}$

the stable manifold drive the action coordinates back to the stable stationary manifold. In Fig. 1b the stable manifold $\mathcal{M}_0$ is colored in green, while the unstable manifold $\mathcal{M}_1$ is colored in red. In the adiabatic limit $\omega_{\text{drive}} \to 0$ kinetic energy terms and dissipative loss terms become negligible and the response of the system follows a path on the stable stationary manifold $\mathcal{M}_0$ that is independent of the speed of driving as long as we do not hit the fence. When staying inside $\mathcal{M}_0$ the response of a system to periodic driving is solely determined by the topology of the stable stationary manifold. If we drive the system across the fence, the action coordinates become unstable. Irreversible processes independent of the driving coordinates $\mathbf{H}_\mathcal{C}$ then let the action coordinates $\mathbf{x}_\mathcal{A}$ leave the stationary manifold and relax back via $\mathcal{C} \otimes \mathcal{A}$ into the interior of the stable stationary manifold $\mathcal{M}_0$.

### 2.3 Projection of paths on the stable manifold into loops in control space

Let $\pi_\mathcal{C} : \mathcal{C} \otimes \mathcal{A} \to \mathcal{C}$ with $\pi_\mathcal{C}(\mathbf{H}_\mathcal{C}, \mathbf{x}_\mathcal{A}) = \mathbf{H}_\mathcal{C}$ denote the projection from product space into control space and $\pi_\mathcal{A} : \mathcal{C} \otimes \mathcal{A} \to \mathcal{A}$ with $\pi_\mathcal{A}(\mathbf{H}_\mathcal{C}, \mathbf{x}_\mathcal{A}) = \mathbf{x}_\mathcal{A}$ denote the projection from product space into action space. Suppose there is a two fold fix point free symmetry operation $S_\mathcal{A}$ ($S_\mathcal{A}^2 = \mathbb{1}$) acting on the coordinates in the action space such that $U(\mathbf{H}_\mathcal{C}, \mathbf{x}_\mathcal{A}) = U(\mathbf{H}_\mathcal{C}, S_\mathcal{A}\mathbf{x}_\mathcal{A})$ and $S_\mathcal{A}\mathbf{x}_\mathcal{A} \neq \mathbf{x}_\mathcal{A}$ for all points $\mathbf{x}_\mathcal{A} \in \mathcal{A}$, then the potential is invariant under the symmetry operation. If the stationary manifold $\mathcal{M}_0$ is path-connected we can choose a path $\gamma \subset \mathcal{M}_0$ from any point $(\mathbf{H}_\mathcal{C}, \mathbf{x}_\mathcal{A}) \in \mathcal{M}_0$ to its symmetry partner point $(\mathbf{H}_\mathcal{C}, S_\mathcal{A}\mathbf{x}_\mathcal{A}) \in \mathcal{M}_0$ and concatenate it with its symmetry partner path $S_\mathcal{A}\gamma$ to form a loop (a closed path) $\mathcal{L}_\mathcal{M} = (S_\mathcal{A}\gamma) * \gamma \subset \mathcal{M}_0$ (here $*$ denotes the concatenation of two paths see Fig. 1b). We note that $\gamma$ is a path but not a loop (magenta in Fig. 1b). Because of the symmetry $S_\mathcal{A}$ in contrast to the path $\gamma$ its projection $\mathcal{L}_\mathcal{C} = \pi_\mathcal{C}(\gamma) = \pi_\mathcal{C}(S_\mathcal{A}\gamma)$ into control space $\mathcal{C}$ is a loop. The projection $\mathcal{L}_\mathcal{C}^2 = \mathcal{L}_\mathcal{C} * \mathcal{L}_\mathcal{C} = \pi_\mathcal{C}(\mathcal{L}_\mathcal{M})$ is a loop circulating twice along the same path in $\mathcal{C}$ and causing a closed loop response $\mathcal{L}_\mathcal{A} = \pi_\mathcal{A}(\mathcal{L}_\mathcal{M})$ that closes only after the second circulation of $\mathcal{L}_\mathcal{C}$. The requirement that there is no fixed point $\mathbf{x}_\mathcal{A}^*$ for which $S_\mathcal{A}\mathbf{x}_\mathcal{A}^* = \mathbf{x}_\mathcal{A}^*$ may be only fulfilled if $\mathcal{A}$ is not a simply connected space (in the example of Fig. 1c $\mathcal{A}$ is a circle). The loop $\mathcal{L}_\mathcal{A}$ thus circulates around a hole of $\mathcal{A}$. For this reason also its preimage $\mathcal{L}_\mathcal{M}$ must circulate around a hole in $\mathcal{M}_0$. If $\mathcal{C}$ is a simply connected space (in Fig. 1a $\mathcal{C}$ is the surface of a sphere and thus simply connected) then $\mathcal{L}_\mathcal{C}^2 = \pi_\mathcal{C}(\mathcal{L}_\mathcal{M})$ must circulate around something other than a hole. In fact $\mathcal{L}_\mathcal{C}$ must circulate around the cusps $\mathcal{B}_\mathcal{C}$ of the fence $\mathcal{F}_\mathcal{C} = \pi_\mathcal{C}(\mathcal{F}_\mathcal{M})$. The cusps $\mathcal{B}_\mathcal{C}$ (bifurcation points) of the fence $\mathcal{F}_\mathcal{C}$ are points in $\mathcal{C}$ where the components of tangent vector $\mathbf{t} = (\mathbf{t}_\mathcal{C}, \mathbf{t}_\mathcal{A}) = (\mathbf{0}, \mathbf{t}_\mathcal{A})$ to the fence $\mathcal{F}_\mathcal{M}$ in the tangent control space vanishes. The punctured control space $\mathcal{C}/\mathcal{B}_\mathcal{C}$ is no longer simply connected and any loop $\mathcal{L}_\mathcal{C} \subset \mathcal{C}/\mathcal{B}_\mathcal{C}$ with non vanishing winding number around one of the cusps $\mathbf{H}_\mathcal{C}^\mathcal{B} \in \mathcal{B}_\mathcal{C} \subset \mathcal{F}_\mathcal{C} \subset \mathcal{C}$ of the fence, causes a half frequency adiabatic response loop $\mathcal{L}_\mathcal{A}$ in action space.

In Fig. 1, we depict the three spaces $\mathcal{C}$, a deformed version of $\mathcal{C} \otimes \mathcal{A}$ with a deformed but topologically equivalent version of the stationary manifold $\mathcal{M}$, and $\mathcal{A}$. We draw the different loops and paths in the three spaces to visualize the arguments made in this section. We use these arguments in section 3 to construct a colloidal adiabatic time crystal.

## 3 Mesoscopic system

### 3.1 The colloidal model

Let us use the knowledge from section 2 to suggest an example of a classical adiabatic discrete time crystal. Our model system consists of a two-dimensional mesoscopic hexagonal magnetic pattern made of up- and down-magnetized domains, see Fig. 2. Such patterns can be produced experimentally using e.g. exchange bias films [47, 48]. The pattern, that is covered

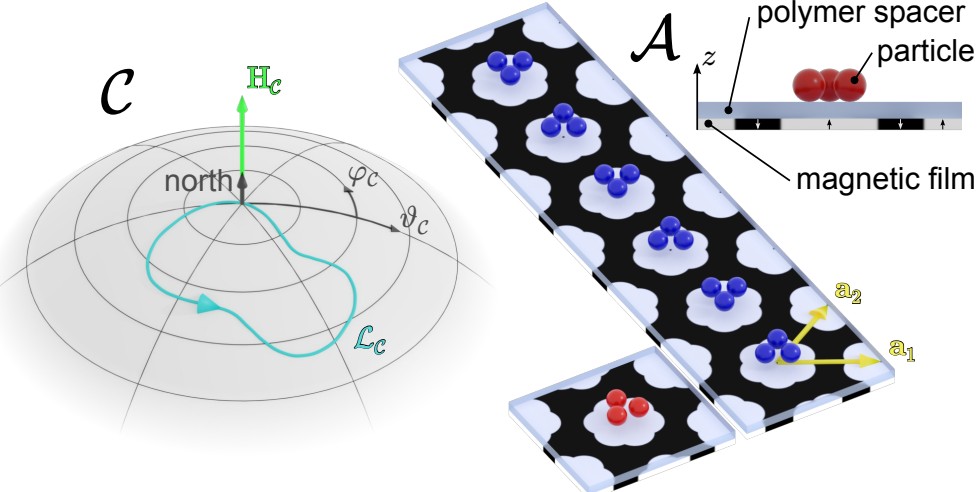

Figure 2: Scheme of the control space $\mathcal{C}$ that is the set of possible orientations of the external magnetic field $H_{\mathcal{C}}$ that we parametrize with the spherical coordinates $\vartheta_{\mathcal{C}}$ and $\varphi_{\mathcal{C}}$ measured with respect to the north direction perpendicular to the magnetic film. Within $\mathcal{C}$ we periodically and adiabatically apply a loop $\mathcal{L}_{\mathcal{C}}$ that we either apply to an ensemble of 3 paramagnetic colloidal particles in one unit cell (red colloids, here shown for a flower shaped domain in the a-conformation) or to 3 paramagnetic colloidal particles per unit cell (blue colloids, here shown for flower shaped domains in the Q-conformation). In both cases the particles are placed on a spacer above a periodic hexagonal magnetic pattern with primitive unit vectors $\mathbf{a}_1$ and $\mathbf{a}_2$ consisting of flower shaped annular up magnetized domains within a counter magnetized surroundings. The equilibrium conformation of the colloids in the flower shaped domains for an external field perpendicular to the magnetic film is an equilateral triangle that can take on two different orientations with corners centered in direction of the lobes of the flower. A time crystalline phase where the colloids respond in a subharmonic way by switching the two orientations after each driving period requires a topologically non-trivial control loop $\mathcal{L}_{\mathcal{C}}$.

with a spacer, creates a two-dimensional magnetic potential that acts on an ensemble of paramagnetic colloidal particles. The colloidal particles sediment due to gravity onto the spacer. An electrostatic levitation from this spacer by roughly the Debye length ensures the mobility of the colloidal particles in action space $\mathcal{A}$. The potential is a function of the positions $\mathbf{x}_{\mathcal{A}} \in \mathcal{A}$ of the particles in action space $\mathcal{A}$, which is the plane parallel to the pattern in which the paramagnetic particles are located. We treat the paramagnetic colloidal particles as being confined to action space. A uniform time dependent external magnetic field $\mathbf{H}_{\mathcal{C}}(t) \in \mathcal{C}$ is also applied to the system. Hence, the total potential depends parametrically on the direction of the superimposed external magnetic field. Our control space $\mathcal{C}$ is a sphere parametrized with coordinates $\vartheta_{\mathcal{C}}$ and $\varphi_{\mathcal{C}}$. In practice we only use the northern polar region of $\mathcal{C}$ where the second Legendre polynomial $P_2(\sin \vartheta_{\mathcal{C}}) < 0$ of the tilt angle of the field is negative and thus lateral dipole-dipole interactions are always repulsive. The paramagnetic particles move in action space $\mathcal{A}$ when we adiabatically modulate the total potential by changing the direction of the uniform external field in control space.

Our two dimensional magnetic hexagonal lattice with primitive unit vectors $\mathbf{a}_1$ and $\mathbf{a}_2$ is built from an arrangement of up and down magnetized domains of a magnetic film. The primitive unit cell of the lattice is a six fold symmetric $C_6$ hexagon containing a down magnetized

matrix that is interrupted by a flower shaped annular up magnetized domain, see Fig. 2. The flower shaped annular domain respects the $C_6$ symmetry of the Wigner Seitz cell, but breaks the continuous rotation symmetry around the central down magnetized domain. We use two different orientations of the flower shaped annulus, that are related by rotating the flower by $2\pi/12$, while keeping the unit cell fixed. In the $a$-conformation the lobes of the flower are located in direction of the primitive unit vectors of the lattice, whereas in the $Q$-conformation the lobes of the flower are located in direction of the primitive reciprocal unit vectors of the reciprocal lattice.

The total magnetic field is the sum of the pattern $\mathbf{H}_p$ and the external $\mathbf{H}_C$ contributions. The potential energy of one paramagnetic particle in the total magnetic field $\mathbf{H} = \mathbf{H}_p + \mathbf{H}_C$ assumes the form $U \propto -\mathbf{H}_C(t) \cdot \mathbf{H}_p(\mathbf{x}_A, z)$ [30] if we apply an external field larger than the pattern field $H_C(t) \gg H_p(\mathbf{x}_A, z)$. For elevations of the particles above the pattern larger than the modulus $a$ of the primitive unit vector, i.e. $z > a$, the potential assumes a universal shape independent of the elevation and independent of the shape of the up magnetized domain. The purpose of the spacer (Fig. 2) is thus to render the potential close to universal such that only the symmetry and not the fine details of the pattern are important. We therefore distinguish thick spacers that render the potential universal from thinner spacers. The case of the flower shaped domain for thinner spacers in the Q-conformation topologically deviates from the universal high elevation case. The behavior in the Q-conformation thus is more subtle and will be discussed in subsection 5.3.

## 3.2 Brownian dynamics simulations

We use Brownian dynamics to simulate the dynamics of paramagnetic colloidal particles. The particles are subject to the single particle potential $U$ from the interference of the external magnetic field with that of the magnetic pattern and they are coupled via dipolar interactions with a pair potential with magnetic moments enslaved to the external magnetic field $\mathbf{H}_C$ (for quantitative details see appendix A). We use inertia free over-damped equations of motion that include a friction force proportional to the particle velocity together with a random force that fulfills the fluctuation dissipation theorem. The particles are confined to the two dimensional action space and the integration of the equations of motion is done using a simple Euler algorithm.

# 4 Breaking the time translational symmetry

## 4.1 Conformations of three particles in a unit cell

In the simulations we place three paramagnetic particles on top of the spacer above the flower shaped annulus domain. The paramagnetic particles are coupled via strong repulsive dipolar interactions. We then switch on a time dependent homogeneous external field $\mathbf{H}_C(t)$.

When the external field points north $\mathbf{H}_C = H_C \mathbf{e}_z$ the single particle potential has a six fold symmetry. The competition of the single particle potential with the repulsive dipolar interactions lets the paramagnetic particles position themselves in form of an equilateral triangle with corners in the direction of every second lobe of the flower shaped domain (see Fig. 3a). Because of the $C_6$-symmetry of the pattern there are two conformational choices for the orientation of the equilateral triangle. One stable conformation is obtained from the other by a $C_6$-rotation operation. Hence the symmetry operation in action space introduced in section 2 for the particular model of section 3 takes on the form of a $S_A = C_6$-rotation. The conformations obtained by a $C_{12}$-rotation of the stable conformation results in another stationary but unstable conformation in which the particles occupy a saddle point of the total energy.

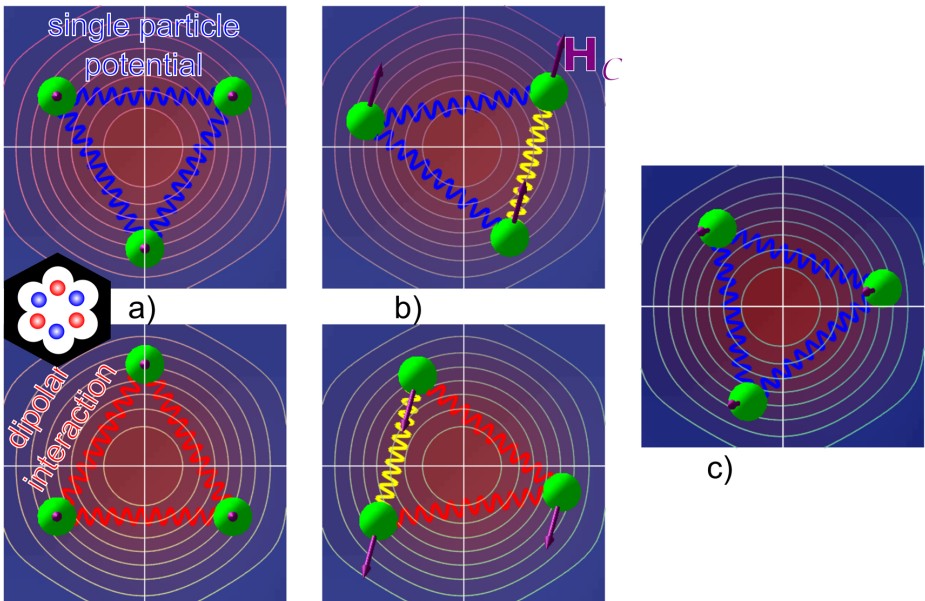

Figure 3: The competition of the single particle pattern potential and the dipole-dipole interaction can lock the orientation of the three particle conformation to the lobes of the single particle pattern potential **a)**, or lock the edge of the triangle of three spheres that experience a reduced dipolar interaction (yellow) to the orientation of the lateral component of the external field **b)**. For both forms of locking there are two possible conformations. One conformation is converted to the other via an inversion operation at the center of the domain. **c)** The third form of locking is with the triangle corner opposite to the lateral component of the external field $H_C$. This single conformation has no symmetry partner conformation and it is favored for high tilt angles of the field and for weak dipolar interactions.

## 4.2 Different ways of locking the orientation of three particles

The dipolar interaction between paramagnetic particles is highly anisotropic. This becomes apparent in Fig. 3b when we redirect the external field $\mathbf{H}_C$ by tilting it slightly from the north pole. The repulsion between paramagnetic particles is then reduced when two interacting paramagnetic particles of the triangle are separated in a direction of the in plane component of $\mathbf{H}_C$. The three paramagnetic particles thus must find a compromise conformation that is either a preference to orient themselves with respect to the single particle potential as shown in Fig. 3a, or to orient two of the three colloidal particles with respect to the in plane external field direction, see Fig. 3b. The preference for the single particle potential becomes stronger when the tilt of the field is small. In such situation the triangle orientation remains locked to the single particle potential as we turn the azimuth of the external magnetic field. Contrary to this, for high tilt angles one edge of the triangle remains locked to the in plane external field direction as we turn the azimuth of the magnetic field. We thus find two topologically distinct classes (cyan respectively yellow/magenta) of loops in control space as shown in Fig. 1a. Both classes of loops consist of starting with an external magnetic field direction at the north pole, tilting the field at constant azimuth, then rotating the azimuth at fixed tilt and returning to the north pole by reducing the tilt at fixed azimuth. The determination to which class the loop belongs is made by the winding numbers of the loop around the six bifurcation points $\mathbf{H}_C^{\mathcal{B}} \in \mathcal{B}_C = \mathcal{F}_C \subset \mathcal{C}$ that for strong dipolar interaction are identical to the fences in control space and thus are a Goldstone mode. The position of the bifurcation points changes with the dipolar interaction strength and with the elevation.

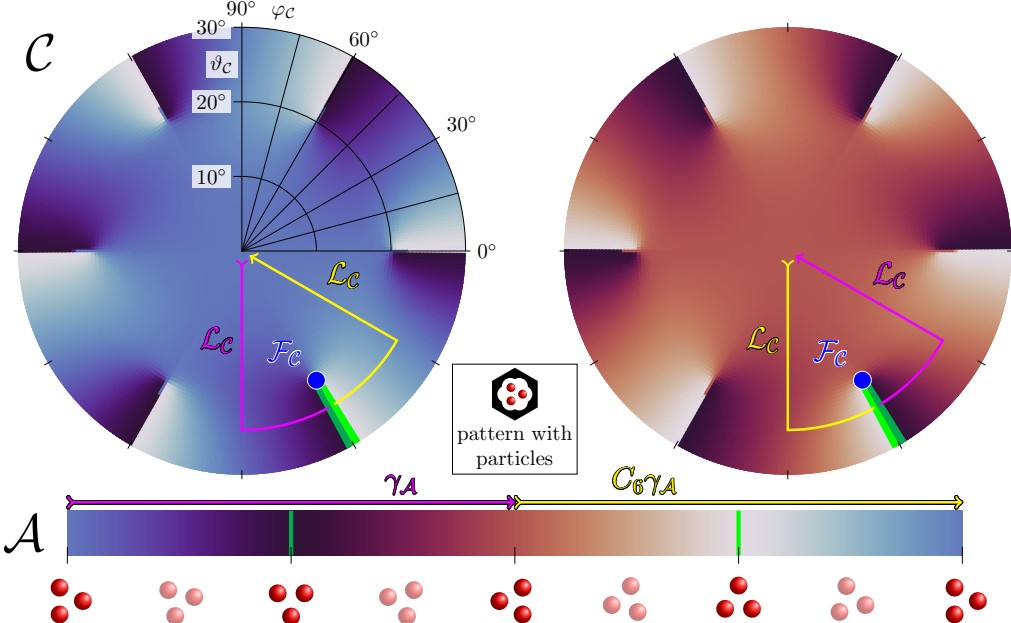

Figure 4: Color-coded equilibrium orientation of the triangle of paramagnetic colloids in action space as a function of the external field orientation in $\mathcal{C}$. Circulating the (blue) fence point $\mathcal{F}_\mathcal{C}$ twice with a purple and yellow loop $\mathcal{L}_\mathcal{C}^2$ causes the subharmonic time crystalline loop $\mathcal{L}_\mathcal{A} = \gamma_\mathcal{A} * C_6 \gamma_\mathcal{A}$ in action space $\mathcal{A}$ shown in the lower part of the figure together with the three particle conformation. Highlighted particle configurations are stationary for a magnetic field pointing north. Those at the beginning and at the end of $\gamma_\mathcal{A}$ or $C_6 \gamma_\mathcal{A}$ are stable for the magnetic field pointing north. Those stable at the magnetic field of the cut would be unstable for the magnetic field pointing north. Two video clips of the trivial and non-trivial dynamics are provided with the videos *adFig4trivial* and *adFig4nontrivial*.

The continuous symmetry of the Goldstone mode becomes broken when dipolar interactions are much weaker than the single particle potential. The potential has a $C_6$ symmetry only for the external field $H_\mathcal{C}$ pointing north. For weak dipolar interactions the symmetry broken parts of the single particle potential at finite tilt of the external field $H_\mathcal{C}$ dominate near the bifurcation point. What has been a fence point develops into a fence line surrounding the region around the bifurcation point in control space. Inside the fence it is energetically advantageous to orient the triangle with a corner antiparallel to the external field, see Fig. 3c. In contrast to the former conformations there is no symmetry partner conformation to this third conformation. Entering such a fenced region kills all attempts to construct a time crystal. The dipolar interaction therefore must be sufficiently strong to enable us constructing a time crystal.

## 4.3 Topologically trivial and non-trivial loops

For the rest of section 4 we assume that the dipolar interaction is strong. A loop not winding around a bifurcation point is a trivial loop causing the conformation to respond with the same period as the driving loop. In contrast a loop winding around a bifurcation point adiabatically connects one stable triangle conformation to the distinct orientation rotated by $2\pi/6$ and the conformation of the paramagnetic particles responds with half the frequency of the frequency in control space.

## 4.4 A three body time crystal

In Figure 4 we show the color coded equilibrium orientation $\varphi_{\mathcal{A}}(\varphi_{\mathcal{C}}, \vartheta_{\mathcal{C}})$ of the triangle of the paramagnetic colloids in action space $\mathcal{A}$ as a function of the external field orientation, expressed with the spherical angular coordinates $(\varphi_{\mathcal{C}}, \vartheta_{\mathcal{C}})$ in control space. We define the orientation angle $\varphi_{\mathcal{A}} = \left(\arccos \frac{a_1 \cdot (2r_3 - r_2 - r_1)}{a |2r_3 - r_2 - r_1|} \mod 2\pi/3\right)$ as the angle modulo $2\pi/3$ between the vector from the triangle center to the corner $r_3$ farthest from the triangle center and the primitive unit vector $a_1$ of the periodic lattice. The corners closer to the triangle center are $r_1$ and $r_2$. The parameters of the simulations are as listed in table 1 of appendix A. The triangle orientation $\varphi_{\mathcal{A}}(\varphi_{\mathcal{C}}, \vartheta_{\mathcal{C}})$ is a double valued function and there are two leaflets of orientations. Both leaflets are glued together at the light/dark green cuts where the orientation is continuous when switching the leaflet at the cut. The non-trivial purple loop $\mathcal{L}_{\mathcal{C}}$ starts on the first blue leaflet and switches onto the second brown leaflet where it returns to the original external field orientation while the orientation $\varphi_{\mathcal{A}}$ follows the open path $\gamma_{\mathcal{A}}$ to a different orientation than at the beginning of $\mathcal{L}_{\mathcal{C}}$, the second yellow revolution of $\mathcal{L}_{\mathcal{C}}$ concatenates the path $\gamma_{\mathcal{A}}$ in $\mathcal{A}$ with $C_6 \gamma_{\mathcal{A}}$ such that the concatenation $\mathcal{L}_{\mathcal{A}} = \gamma_{\mathcal{A}} * C_6 \gamma_{\mathcal{A}}$ is a subharmonic loop causing a discrete time crystalline response. The non-trivial behavior occurs whenever one circulates around the fence $\mathcal{F}_{\mathcal{C}}$ in control space. For trivial control loops with vanishing winding numbers around each of the six fence points, the response in action space remains trivial, causing no time crystalline behavior. On the arxiv we provide two videos (*adFig4trivial* and *adFig4nontrivial* [2]) of Brownian dynamics simulations of trivial and non-trivial control loops together with their trivial and non-trivial response. Albeit being stored on the arXiv these videos are an integral part of this work.

An external field that points into the direction of a bifurcation point may be viewed as a loop of infinitesimal radius around the bifurcation point, for which each possible orientation is taken. The orientational degree of freedom becomes a Goldstone mode [49] at the bifurcation point, where it obeys a generalized statistical mechanics Noether theorem [50]. As already mentioned the continuous symmetry of the Goldstone mode becomes broken when dipolar interactions are much weaker than the single particle potential.

## 5 Breaking the space and time translational symmetry

The control loops of the homogeneous external field in control space of section 4 had a discrete time translational symmetry that was broken by the dynamics of the paramagnetic particles. If we arrange several magnetic annular domains into a periodic pattern, the magnetic field of the potential has a discrete translational symmetry in space. In this section we use Brownian dynamics simulations to show that there are ways to break or not break some or all of the discrete symmetries, once we fill some of the neighboring flower shaped annular domains with three paramagnetic particles in a periodic manner. We use two different orientations of the flower shaped annulus, that are related by rotating the flower by $2\pi/12$, while keeping the unit cell fixed (see Fig. 5). In the $a$-conformation the lobes of the flower are located in direction of the primitive unit vectors of the lattice, in the $Q$-conformation the lobes of the flower are located in direction of the primitive reciprocal unit vectors of the reciprocal lattice. The dipolar interaction is long range and anisotropic and there are intercellular interactions between the paramagnetic particles of neighboring annular domains. We have to distinguish the behavior of flowers in the $a$- and in the $Q$-conformation, if working at lower than universal elevations. Note that the differences between both patterns become irrelevant at universal

---

[2]*supplementary videos*, are provided in the ancillary directory https://arxiv.org/src/2203.04063/anc. These videos are essential for understanding the dynamics of our time crystals.

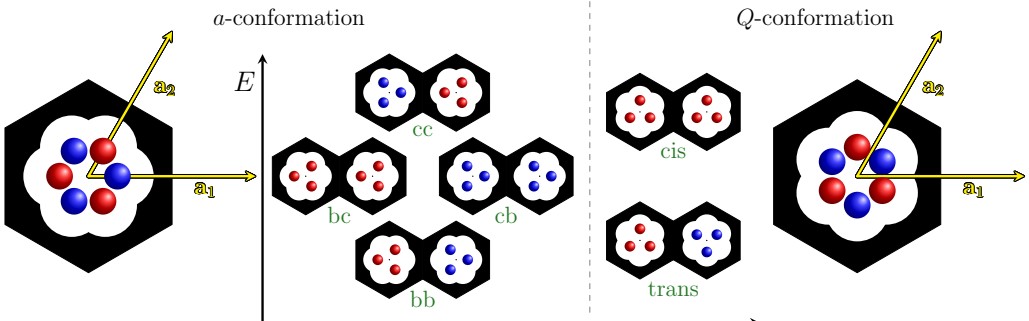

Figure 5: The two configurations of paramagnetic particles in red and blue within a unit cell for an external magnetic field $\mathbf{H}_\mathcal{C}$ pointing north (normal) to the pattern in the a-conformation and in the Q-conformation. Intercellular dipolar interactions lead to a splitting of the energy. We distinguish $cc$, $cb$, $bc$, and $bb$ bonds in the a-conformation and $trans$ and $cis$ bonds in the Q-conformation. The energy splittings cause many body interactions that induce various kinds of spatio-temporal order.

elevations. We can continuously move from the $a$- toward the $Q$-conformation by joining both conformations at a universal height. In subsections 5.2 and 5.3 we will show that the $a$-conformation suppresses the space-time crystalline behavior in the bulk, but not the time crystalline response at the spatial edge of the crystal, while the $Q$-conformation supports a bulk-space-time crystalline response.

## 5.1 Space-time crystalline order

The magnetic pattern has primitive unit vectors $\mathbf{a}$ and primitive reciprocal unit vectors $\mathbf{Q}$. Together with the period $T = 2\pi/\omega_{\text{drive}}$ of the external magnetic field loop in control space, we can define 4-vectors $\mathbf{r}^4 = (t, \mathbf{r})$ and primitive unit vectors $\mathbf{a}^4$ in space-time. A trivial response of the paramagnetic particles is when the reciprocal (angular frequency, wave vector) lattice of the response is the same as the reciprocal lattice of the drive $(\omega_{\text{response}}, \mathbf{Q}_{\text{response}}) = (\omega_{\text{drive}}, \mathbf{Q}_{\text{drive}})$. Depending on the strength of the intra- and inter-cellular dipolar interactions, on the conformation of the flower shape annular domain, and on the elevation of the paramagnetic particles above the pattern we find different disordered phases. For example we find orientation disordered time crystalline phases, for which the orientation order parameter $O = \cos(3\varphi_\mathcal{A})$ between neighboring unit cells is completely uncorrelated $\langle O(\mathbf{r}^4 + \mathbf{a}^4)O(\mathbf{r}^4)\rangle = 0$ for any primitive unit vector $\mathbf{a}^4$ of the space-time lattice. Here $\varphi_\mathcal{A}$ denotes the orientation angle of one of the three particles in a unit cell. We find frozen disordered phases, for which the orientation order parameter between neighboring unit cells is uncorrelated $\langle O(\mathbf{r}^4 + \mathbf{a}_D^4)O(\mathbf{r}^4)\rangle = 0$ for the disordered primitive unit vector $\mathbf{a}_D^4$ direction but completely correlated (anticorrelated) $\langle O(\mathbf{r}^4 + \mathbf{a}_F^4)O(\mathbf{r}^4)\rangle = \pm 1$ along the frozen lattice direction $\mathbf{a}_F^4$ of the space-time lattice. The time translational symmetry of those frozen disordrered phases is not broken if all primitive unit vectors, having a non-vanishing time component, are positively correlated frozen primitive unit vectors. These phases thus are not discrete time crystalline phases. In any other case the order is either a disordered time crystalline frozen space or a disordered space sub harmonic time crystalline order.

We also find completely ordered phases that are correlated (anticorrelated) in any of the space-time lattice directions. If the primitive unit vectors can be separated into primitive unit vectors along time and primitive unit vectors along space, we find non time crystals for which the order parameter correlation along the time direction is positive. Depending on the correlation in space we find positively correlated order of neighboring cells in which case the order is

*ferromagnetic* and neither the time nor the space translational symmetry is broken, or we find at least one anticorrelated space direction in which case the order is *antiferromagnetic* and the space translational symmetry is broken with an ordered primitive unit cell twice the size of a primitive unit cell of the driving lattice. If the order is antiferromagnetic in the direction of time we get a time crystal again with a unit cell twice the size of the primitive unit cell of the driving lattice. For a spatially ferromagnetic time crystal the primitive time unit vector of the order is double the primitive time unit vector of the drive. For a spatially antiferromagnetic time crystal the primitive unit vectors of the lattice are no longer separable into spatial and time-like unit vectors but they point along the diagonals between space and time similar to a sodium chloride structure in a spatial crystal.

Given the multitude of crystalline structures in space it should not surprise us that we also find a multitude of different more or less complex space-time structures. Next, we will show how to attain some of those structures in our colloidal model system.

## 5.2 Topologically isolating time crystals

The dipolar interaction is long range and anisotropic. Hence we can change the time crystalline behavior by filling unit cells of the pattern in an anisotropic way. We must avoid weak dipolar interaction because under such circumstance the regions where the unique conformation shown in Fig. 3c is adopted connects all bifurcation points and can no longer be circulated. When the dipolar interaction is moderate the inter-cellular interactions do not matter and the colloidal particles in one cell respond independently of those in the other cell. We hence find a time crystal with frozen spatial disorder. For the $a$-conformation at stronger dipolar interactions the degeneracy of the two distinct triangular conformations is broken and we can distinguish three types of bonds between neighboring cells (see Fig. 5). A $cc$-bond with two corners of neighboring triangles facing each other has a higher energy than the lowest energy $bb$-bond where two triangular bases face each other. A $bc$-bond has intermediate energy. Strong dipolar interactions can therefore also destroy the subharmonic response in ensembles of filled cells. The $a$-conformation is therefore not a good conformation for time crystals if all the flower shaped domains are occupied.

We consider fillings of cells such each filled cell has only bonds to filled neighboring cells where the bonds form an angle of $2\pi/3$ or $4\pi/3$. Filled cells form a Kagome lattice (see Fig. 6) consisting of Kagome Wigner Seitz cells built from two filled (green and yellow) unit cells and one empty (gray) unit cell of the magnetic pattern. Under these conditions each filled cell is either a $b$-bonding or $c$-bonding cell exposing the same type of bond toward each of its occupied neighbor cells. When the splitting of bond energies due to intercellular dipole interactions are pronounced, a $bb$-bond cannot be converted into any other bond. This domination occurs for external fields that point in the surroundings of the single cell fence points in control space, where the symmetry of the continuously degenerate Goldstone mode can be easily broken by the intercellular dipole interactions. The inter-cellular dipole interaction perturbed fences in control space thus no longer coincide with the single cell unperturbed fence points but form closed curves $\mathcal{F}_{\mathcal{C}}^{\ i}(z_i)$, that contain bifurcation points and that depend on the type $i = 1, 2$ of the filled cell in the Kagome lattice as well as the number $z_i$ of filled neighboring cells. Inside the fence of one cell only one minimum, namely the $b$-bond orientation of the triangles of the cell, exists and whenever crossing inside to this unique region the space-time crystal no longer responds with half the driving frequency.

For intermediate intercellular dipolar interaction strength, the same loop can enter the interior of the fence for one unit cell but not for another unit cell. In such cases one still finds a triangle in one cell responding with half the driving frequency and the triangle in the other cell responding with a trivial loop. The dynamics of the trivial responding cell is no longer adiabatic: When a $c$-bonding triangle is driven inside the unique $b$-bonding fenced region an

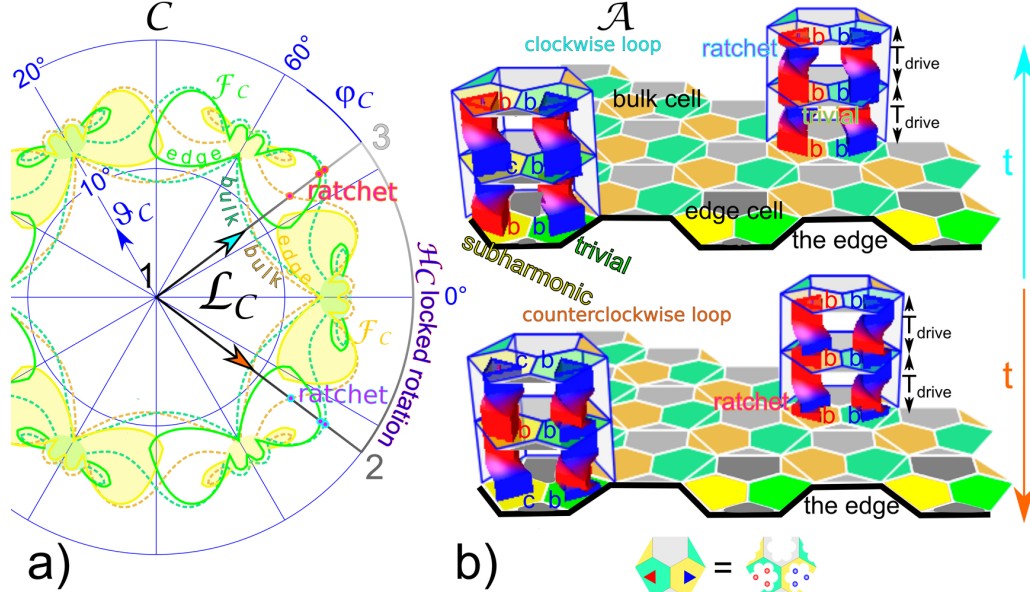

Figure 6: A complex topologically isolated space-time crystal in the a-conformation:
**a)** A counterclockwise (clockwise) loop $\mathcal{L}_\mathcal{C}$ in control space $\mathcal{C}$ circulates the solid
yellow $\mathcal{F}_\mathcal{C}$-edge fences, but cuts through the solid green $\mathcal{F}_\mathcal{C}$ edge-fences as well as
through all dashed bulk fences.     **b)** Snapshot of a space-time crystal of a hexagonal
pattern with a Kagome-lattice of filled cells including the spatial edge of the lattice
at the time when the loop is at the north pole. The space-Wigner-Seitz-cell of the
Kagome lattice consists of two (green and yellow) filled cells and one empty (gray)
cell. At the spatial edge (there is no edge in the direction of time) of the Kagome-
ribbon, filled cells have a reduced $z = 2$ configuration number. The bulk cells as well
as the green edge cells relax into a spatial antiferromagnetic order of only $b$-bonding
cells when the loop returns to the north-pole. The response of the yellow edge cells
is subharmonic in time and the spatial disorder is frozen repeating in time with every
second loop. We have depicted one edge-space-time Wigner-Seitz-cell and one bulk-
space-time Wigner-Seitz-cell onto the Kagome lattice to show the dynamics of the
triangle conformation. The sequence of conformations in the yellow edge cell is time
reversal invariant, while the sequence of conformations in all other cells are not time
reversal invariant. Two space-time-Wigner-Seitz edge cells can be packed side by side
in space or with a symmorphic translation by one lattice vector in space and half a
primitive vector in time creating the frozen disordered topological time crystalline
edge wave in a). On the arXiv two video clips of the dynamics are provided with the
videos *adFig6clockwise* and *adFig6counterclockwise*.

irreversible ratchet jump of the triangle from the no longer stable $c$-bonding toward the $b$-
bonding orientation occurs. The jump cannot be undone by backing up on the loop across the
fence into the twofold orientation region outside the fenced region.

   The response of the conformation of three colloidal particles in a particular unit cell $i$ then
depends on the configuration number $z_i$, i.e. the number of neighboring cells being also filled
with three colloidal particles. The time crystalline topological response is robust for small
configuration numbers $z < 2$. However the response changes for colloidal particles sitting
in a bulk cell $z = 3$ or at an arm-chair edge cell $z = 2$ of a Kagome lattice. For intermediate
dipolar interaction strength $bb$-bonds are converted to $bb$-, $bc$-, and $cb$-bonds rather than into
$cc$-bonds, because only one of the edge cells responds as a subharmonic time crystal, while all

bulk cells and the other edge cell show trivial harmonic responses. Note that at the edge of a Kagome lattice the translation symmetry normal to the edge is explicitly broken, while the translation symmetry in time parallel to the edge is only broken due to the time crystalline response of the colloids inside the lattice. Within the bulk of the filled Kagome-lattice we distinguish 2 different filled ($z = 3$) unit cells depending on the two possible directions of the neighboring cell triple. At an arm-chair edge of the Kagome lattice the configuration number of the outermost two cells reduces to $z = 2$. Each of these two cells have a somewhat different fence in control space for each of their possible configuration number. A triangle subject to a magnetic field $\mathbf{H}_{\mathcal{C}}$ tilted into a certain direction is displaced from the center of a flower in the same direction. If the tilt is toward a neighboring filled cell, then the unique $b$-bonding fenced region is larger than when the tilt is pointing away from the filled cell. For the same region in control space therefore the unique fenced region alternates in size as we move from one to a neighboring cell that has its bonds into the opposite directions. It is then easy to create a time crystal where e.g. only one of the edge cells has subharmonic response while the other cells respond in a trivial irreversible way. We can use these results to form an irreversible, i.e non-time reversal invariant, non-time bulk crystal, with time-crystalline edge states where the response in time and in space is quite complex.

In Fig. 6 we show the response of Kagome-lattice ribbon of two filled (green and yellow) cells and one empty (gray) cell to a control loop $\mathcal{L}_{\mathcal{C}}$ that consist of three sections. The control loop starts at the north pole (position 1) and moves at constant azimuth $\varphi_{\mathcal{C}}^i$ (measured with respect to the $\mathbf{a}_1$ direction) toward the tilt angle $\vartheta_{\mathcal{C}}^{max}$ (position 2) where it turns to the final azimuth $\varphi_{\mathcal{C}}^f$ (position 3) before it returns to position 1 (the north pole). The values of $\varphi_{\mathcal{C}}^i$, $\varphi_{\mathcal{C}}^f$, and $\vartheta_{\mathcal{C}}^{max}$ are chosen (simulation parameters see table 1 in appendix A) such that the loop circulates around the (yellow) numerically computed $\mathcal{F}_{\mathcal{C}}$-edge-fences, but cutting through the green $\mathcal{F}_{\mathcal{C}}$- edge-fences with the 12, and 31-sections of the loop. The loop also cuts through all the bulk fences but avoids unique regions inside the yellow edge-fences causing the adiabatic subharmonic response of the yellow edge cells, The triangles of paramagnetic particles in all bulk cells and in the green edge cells, however, perform a ratchet jump from a $c$-orientation toward a $b$-orientation whenever the particles in these cells are in the unstable $c$-orientation prior to the fence passing segment resulting in a trivial harmonic response to the loop. The irreversibility of the dynamics in the bulk cells and in the green edge cells becomes apparent when we reverse the driving loop causing the irreversible triangles of the time crystal to follow a sequence of orientations that is not the reversed sequence of the forward loop $\mathcal{L}_{\mathcal{A}}(\mathcal{L}_{\mathcal{C}}^{-2}) \neq (\mathcal{L}_{\mathcal{A}}(\mathcal{L}_{\mathcal{C}}^2))^{-1}$. Adiabatic response is time reversal invariant, while irreversible non-adiabatic motion is not. On the arXiv we show with the videos *adFig6clockwise* and *adFig6counterclockwise* the full dynamics of the topological insulating time crystal shown in Fig. 6 for the counterclockwise $\mathcal{L}_{\mathcal{C}}$ as well as for the clockwise $\mathcal{L}_{\mathcal{C}}^{-1}$ driving loops.

The behavior of the irreversible cells is synchronized for all cells. Bulk cells and the green edge cells are all $b$-bonding when the loop returns to the north pole. The adiabatic (yellow) edge cells follow a subharmonic response whatever the initial order of the adiabatic cells were in the beginning. The adiabatic cells therefore are generically spatially disordered but time crystalline frozen as a function of time. The trivial edge cells and all bulk cells are spatially ordered in an antiferromagnetic order. It is an antiferromagnetic crystal with topologically protected time crystalline edge states.

## 5.3 Time crystalline phase transitions

In the $Q$-conformation at lower than universal elevation paramagnetic colloidal particles in neighboring unit cells can be arranged in either the trans-conformation (bond direction crossing the primitive unit vector connecting the neighboring cells) or in the cis-configuration (bond

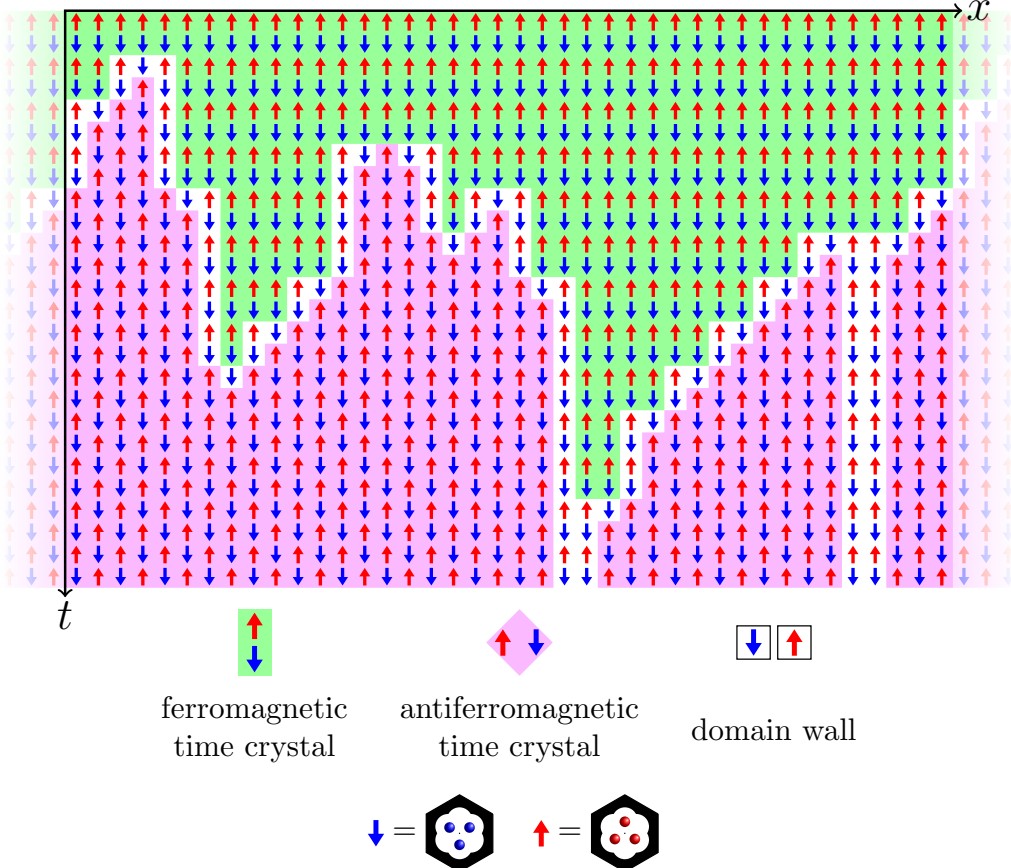

Figure 7: Stroboscopic development of a chain of honeycomb cells filled with three colloidal particles each. The initially ferromagnetic ordered time crystal undergoes a topological transition towards an antiferromagnetic time crystal. Space-time Wigner Seitz cells of the various orders are shown at the bottom. The spatial order in each cell is abbreviated by a blue or red arrow as indicated. A video clip of the dynamics is provided with the video *adFig7*

.

parallel to the cell connecting primitive unit vector) see Fig. 5. The trans-conformation has a lower energy than the cis-conformation. Hence, an antiferromagnetic ordering of the triangles along a row of unit cells is preferred. Neither a ferromagnetic nor an antiferromagnetic ordering suppress the time crystalline subharmonic behavior since in contrast to the behavior in the previous section, intercellular bonds between colloidal particles before and after an adiabatic driving cycle remain equivalent. In Fig. 7 we show the evolution of a ferromagnetic time crystal at time zero towards an antiferromagnetic time crystal for the simulation parameters listed in table 1 of appendix A. One row of neighboring cells along the $\mathbf{a}_1$-direction are filled with three particles each. The originally ferromagnetic time crystal remains in the ferromagnetic time crystalline order for a typical correlation time until a few antiferromagnetic time crystalline nuclei are formed. These nuclei grow and two more stable antiferromagnetic time crystalline ordered phases (of the same order but connected via a non-symmorphic translation that is part of the unbroken but not of the broken symmetry) separated by space like domain walls appear. They thus replace the old order with the new antiferromagnetic time crystalline order. The transition from the ferromagnetic towards the antiferromagnetic ordering is irreversible. In contrast to the time crystalline order in the a-conformation the space-time-bulk of the antiferromagnetic time crystal shows fully adiabatic response, and a forward or backward

loop $\mathcal{L}_{\mathcal{C}}$ or $\mathcal{L}_{\mathcal{C}}^{-1}$ results in a sequence of orientations that are the time reversed version of the other. A video clip of the full dynamics is shown in the video *adFig7*.

The dipolar interaction couples the particles and broadens the one dimensional fence $\mathcal{F}_{\mathcal{C}}$ to a two dimensional object of control space. Here the intercellular dipolar interactions are strong enough to destroy the path connection of the ferromagnetic and antiferromagnetic conformations on $\mathcal{M}_0$. We can hence distinguish the fences $\mathcal{F}_{\mathcal{M}_0^{a,f}}$ of the antiferromagnetic and the ferromagnetic stable conformations and the path disconnected subsets $\mathcal{M}_0^{a,f}$ of the stable stationary manifolds enclosed by those fences. The only way to dynamically move from the ferromagnetic region $\mathcal{M}_0^f$ toward the antiferromagnetic region $\mathcal{M}_0^a$ is by driving the system toward the ferromagnetic fence $\mathcal{F}_{\mathcal{C}}{}^f = \pi_{\mathcal{C}}(\mathcal{F}_{\mathcal{M}_0^f})$ where a ratchet jump from $\mathcal{F}_{\mathcal{M}_0^f}$ via $\mathcal{C} \otimes \mathcal{A}$ into the lower potential subsection $\mathcal{M}_0^a$ occurs. In our simulations the control loop $\mathcal{L}_{\mathcal{C}}$ circles around the ferromagnetic fence $\mathcal{F}_{\mathcal{C}}{}^f$ by almost touching it such that the thermal fluctuation forces eventually drive the system across the fence. The persistence time of the ferromagnetic phase therefore sensitively depends on the proximity of the loop to the ferromagnetic fence as well as on the never truly adiabatic speed in passing this sensitive section of the loop. Once in the antiferromagnetic time crystalline phase one cannot return to the ferromagnetic conformation because there is no point of the antiferromagnetic fence that has a higher potential than the corresponding point of the ferromagnetic stable manifold $\mathcal{M}_0^f$. The antiferromagnetic phase therefore persists.

## 6 Conclusion

We have shown that the topology of the stationary manifold embedded in the product space of the external control variables and the quasistatic response variables determines whether a time crystalline driving is possible or not. Small thermal or quantum fluctuations will not change the topological phenomena since these are robust against weak perturbations. As long as transition or tunneling rates between the distinct minima are tiny, the time crystalline behavior will prevail also for a quantized version of the model. The essential requirement for the adiabatic time crystal to work is the non-ergodicity, i.e. the population of only one of the well separated minima. For these type of topological time crystals the quantum respectively classical nature of the phenomenon seems to be of minor importance. On a mesoscopic scale, proper cooling, necessary for any motor, whether in a quasi-equilibrated reversible cycle or driven far from thermal equilibrium is not a problem such that over heating [51–56] of the time crystal can be prevented. A rich variety of topologically induced non-time crystalline and time crystalline phases can be found both in the bulk or at the edge when playing with the competition of intra- and intercellular dipolar interactions between the paramagnetic colloidal particles of the many body ensemble.

## Acknowledgements

We thank Daniel de las Heras for scientific discussion, critical comments and help.

**Funding information**   We acknowledge funding by the Deutsche Forschungsgemeinschaft (DFG, German Research Foundation) under project number 440764520.

# A  Numerical details and parameters

The pattern magnetic field at the elevation $z$ above the pattern reads

$$\mathbf{H}_p(\mathbf{x}_{\mathcal{A}}, z) = \int \frac{(\mathbf{x}_{\mathcal{A}} + z\mathbf{e}_z - \mathbf{x}'_{\mathcal{A}})}{4\pi((\mathbf{x}_{\mathcal{A}} - \mathbf{x}'_{\mathcal{A}})^2 + z^2)^{3/2}} M_z(\mathbf{x}'_{\mathcal{A}}) d^2\mathbf{x}'_{\mathcal{A}}, \tag{2}$$

where

$$M_z(\mathbf{x}_{\mathcal{A}})\mathbf{e}_z = \begin{cases} +M_s\mathbf{e}_z & \text{if } |\mathbf{x}_{\mathcal{A}} - \mathbf{a} - r_M\hat{\mathbf{e}}_i| < r_K \\ & \text{for one lattice vector } \mathbf{a} \\ & \text{and one normed primitive} \\ & \text{lattice vector } \hat{\mathbf{e}}_i \ (i = 1..6) \\ \\ -M_s\mathbf{e}_z & \text{else} \end{cases}, \tag{3}$$

is the magnetization of the thin magnetic film with $M_s$ the saturation magnetization, $r_M = 0.2a$, $r_K = 0.19a$, and $\hat{\mathbf{e}}_i = \mathbf{a}_i/a\,(\mathbf{Q}_i/Q)$ are normed vectors in the direction of the primitive unit vectors of the direct (reciprocal) lattice for the flower domains in the a-conformation (Q-conformation).

The single particle potential is

$$U = -\mu_0 \mathbf{H}_p(\mathbf{x}_{\mathcal{A}}, z) \cdot \mathbf{m}, \tag{4}$$

with $\mathbf{m} = \chi_{eff} V \mathbf{H}_C$ the magnetic moment of the paramagnetic particle with volume $V$ and effective magnetic susceptibility $\chi_{eff}$. The dipolar interaction reads

$$U_{dipol} = \frac{\mu_0}{4\pi} \frac{(\mathbf{x}_{\mathcal{A}} - \mathbf{x}'_{\mathcal{A}})^2 \mathbf{m}^2 - 3((\mathbf{x}_{\mathcal{A}} - \mathbf{x}'_{\mathcal{A}}) \cdot \mathbf{m})^2}{|\mathbf{x}_{\mathcal{A}} - \mathbf{x}'_{\mathcal{A}}|^5}. \tag{5}$$

We use dimensionless units of length normalized to the lattice constant $a$, of the magnetic field normalized to the effectively attenuated magnetization $M_{eff} = \gamma(\mathbf{Q}_1)M_s e^{-Qz}$ of a fictive universal pattern at elevation $z$ with $M_s$ the saturation magnetization of the real pattern, $Q = 4\pi/\sqrt{3}a$ the modulus of the primitive reciprocal unit vectors, and $\gamma(\mathbf{Q}_1) = \int_{WZ} e^{i\mathbf{Q}_1 \cdot \mathbf{x}_{\mathcal{A}}} M_z(\mathbf{x}_{\mathcal{A}}) d^2\mathbf{x}_{\mathcal{A}}/M_s \frac{\sqrt{3}}{2}a^2 \approx 0.3$ the leading reciprocal lattice Fourier coefficient of both patterns, where the Fourier integral is taken over the Wigner Seitz cell $WZ$ of each pattern. Units of energy are normalized to $\mu_0 M_s^2 a^3$ and non dimensional effective magnetic susceptibilities $\chi_{eff} V/a^3$. Our control loops start a the north pole and move at constant azimuth $\varphi_C^i$ (measured with respect to the $\mathbf{a}_1$ direction) toward the tilt angle $\vartheta_C^{max}$ where they turn to the final azimuth $\varphi_C^f$ before they return to the north pole. In table 1 we list the parameters used to compute the time crystals shown in Figs. 4, 6, and 7.

Table 1: Simulation parameters

| | $\dfrac{H_C \chi_{eff} V e^{4\pi z/\sqrt{3}a}}{\gamma(\mathbf{Q}_1)M_s a^3}$ | $z/a$ | $\varphi_C^i$ | $\varphi_C^f$ | $\vartheta_C^{max}$ | conformation |
|---|---|---|---|---|---|---|
| Figure 4 | $1.64\,10^{-1}$ | universal | $-90°$ | $-30°$ | $23°$ | a, single cell |
| Figure 6 | $1.09\,10^{-2}$ | universal | $-37°$ | $37°$ | $20°$ | a |
| Figure 7 | $6.5\,10^{-1}$ | $1/3$ | $71.7°$ | $124°$ | $33.2°$ | Q |

All integrations were performed numerically using an Euler algorithm. Loops were followed adiabatically by reducing the speed of modulation to values that do no longer affect the outcome.

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
