# Peer review of "Adiabatic and irreversible classical discrete time crystals"

_SciPost Physics, doi:SciPost Phys. 13, 091 (2022)_

## Round 1 · Referee Report · Anonymous (Referee 1) · 2022-7-1

Strengths

The paper introduces discrete time crystals based on the conformation of colloidal particles above a magnetic hexagonal pattern and subjected to external periodic field. The authors demonstrate complex periodic response of the system and dynamic phase transitions between reversible and non-reversible dynamic particle arrangements defined by the topology of the control loops driving the system. The complexity is further multiplied by the role of long-range multibody interactions and anisotropic filling of the cells.
Colloidal time crystal may introduce a powerful platform for topologically protected transport phenomena in driven colloidal ensembles.

Weaknesses

The paper is not easy to understand due to the complexity of the problem at hand. That is not a real weakness though. The authors tried their best to systematically introduce the required terminology and concepts.

Report

The paper addresses an important aspect of topology in space and time to generate a nontrivial dynamic response of the system on external periodic driving. The paper introduces concepts and approaches using the example of a magnetic colloidal system driven above the patterned magnetic substrates to demonstrate a rich variety of topological time crystalline and non-time crystalline phases phenomena available. I was humbled by the complexity of the system.
The paper is well written and suitable for publication in SciPost.

Requested changes

the paper introduces a lot of variables that are needed for the story, but some of them are not defined in time. For instance, on page 8 when defining \phi_A authors use vector a_1 that is not defined right away and only on page 10.

---

## Round 1 · Referee Report · Anonymous (Referee 2) · 2022-7-10

Strengths

1- Glossy figures
2- Nice videos

Weaknesses

Difficult to follow

Report

The authors propose a realization of classical discrete time crystals in a system of paramagnetic colloidal particles that are placed above a magnetic hexagonal pattern and exposed to an external field that changes its direction periodically.

I have to admit that this work is a bit peripheral to my expertise. Thus, although it does seem interesting, I have confess that I have a bit of a hard time following it.

Part of the issue may be related to the figures that are glossy, but nevertheless difficult to understand.

Figure 1 is particularly difficult to understand. If its description would mention specific parts (left, middle, or right panel), this would probably already help. Likewise, section 2 is quite abstract and thus not easy to read, at least not for a non-expert.

Another detail is recurrent terminology such as "north pole". I only found a short explanation on page 5 during a careful second reading. At the same time, the angles $\vartheta$ and $\varphi$ are not properly introduced. Conversely, line 2 of the caption of Fig. 5 mentions a direction that is "normal to the pattern", which might be identical to "north".

The caption of Fig. 7 should mention that this space-time picture is for a chain of honeycomb cells.

The five accompanying video clips are extremely useful to clarify and illustrate the findings of the present work. However, I believe that it would be good to emphasize that these videos are not just a supplementary piece, but actually an important part of this work. At the same time, the source in Ref. [49] needs to be specified more clearly. It may be Ok to use arXiv for them, but since this is not SciPost, this should be stated explicitly.

Content-wise, the middle panel of Fig. 1 does not factorize into a control space ${\cal C}$ and an action space ${\cal A}$, as the notation ${\cal C} \otimes {\cal A}$ seems to suggest.

A further comment on content is the emphasis on the behavior at the edge in section 5.2. After all, a spatial crystal should break translational symmetry spontaneously while an edge breaks it explicitly. I would be grateful if the authors could clarify this point.

Final remark on content: At the very end of the text, the authors say "All integrations were performed numerically". I think that the authors should give a few more details, and in particular specify the integration algorithm.

I would like to see the above points clarified before I commit on a definite recommendation for SciPost Physics.

There are a few further minor issues that I list among the "Requested Changes".

Requested changes

1- Make the description of Fig. 1 more accessible.
2- Try to explain the content of section 2 better to a non-expert.
3- Define the angles $\vartheta$ and $\varphi$ properly and use this opportunity to specify the meaning of the "north pole".
4- Clarify if the "normal to" on line 2 of the caption of Fig. 5 is synonymous to "north".
5- The caption of Fig. 7 should mention that this space-time picture is for a chain of honeycomb cells.
6- Check if the terminology "nuclei" (several) versus "nucleus" (one) is used correctly in the video clip "adFig7.mp4".
7- Emphasize in the text that the "supplementary videos" are an integral part of this work.
8- Specify the source of in Ref. [49] (arXiv?) more clearly.
9- Clarify the notation ${\cal C} \otimes {\cal A}$, or possibly rather the meaning of ${\cal C} $ and ${\cal A}$, as appropriate.
10- Clarify the relevance of the edge in section 5.2 (Fig. 6(b)): in which sense is the resulting state still a crystal?
11- Expand the comment "All integrations were performed numerically" at the end of the Appendix. In particular, specify the integration algorithm.
12- Page 11: eliminate abbreviations that are not used again ("DO TC") or only a second time ("FDO").
13- Page 11: Using angles $\langle \cdot \rangle$ rather than less and bigger signs $<\cdot>$ for expectation values would improve readability.
14- Still page 11: The "timely" should probably rather be a "time" or "time-like".
15- Caption of Fig. 6: I think that "spatial" would be preferred spelling over "spacial".
16- Format Eq. (A3) such that the curly bracket "$\{$" stretches across both alternatives.
17- Last line of text: "effect" $\to$ "affect".
18- Properly upper-case names in the titles of Refs. [16,18,20,27,29,36,38,56].
19- Correct the spelling of the name of the author of Refs. [52,53]: "T. Prosen".

---

## Round 2 · Referee Report · Anonymous (Referee 1) · 2022-7-25

Report

The authors appropriately addressed all the comments raised by the reviewers. I recommend the manuscript for acceptance.

---

## Round 2 · Referee Report · Anonymous (Referee 3) · 2022-7-26

Report

I had known the manuscript before I got it for reviewing. The response of the authors to all remarks of the previous refferees is fully satisfactory. The manuscript itself is interesting and worth publishing in SciPost. Therefore, I recommend its acceptance for the publication.

---

## Round 2 · Referee Report · Anonymous (Referee 2) · 2022-8-4

Report

The authors have improved their manuscript such that I now have the pleasure to recommend publication in SciPost Physics.

I am aware that it is probably be a bit late, but since a question occurred to me when rereading the manuscript, let me mention it anyway: An essential characteristic of antiferromagnetic (Néel) states is a two-fold degeneracy. Fig. 7 illustrates the case of two compatible domains merging. Nevertheless, I was wondering what would happen if two antiferromagnetic domains with opposite phases met.

Requested changes

A few minor items of a more typographical nature: 1- There may be different versions of the hyphenation rules, but I think that "Debye length", "bulk cells", and "edge cells" should be written without a hyphen. Furthermore, whatever the rules, they should be applied consistently. For example, I have found "inter cellular", "inter-cellular", and "intercellular" in the manuscript ("intercellular" might be the best option, but please unify). 2- Line 9 of caption of Fig. 6: "a" (not "an") before "spatial". 3- Line 4 of page 14: typeset $z=3$ in math mode. 4- Full stop missing at the end of the caption of Fig. 7 and in Eq. (5). 5- There is no need for separate hyperlinks in Refs. [43,48,57] given that there is a DOI. 6- Ref. [47]: SciPost allows footnotes, i.e., no need for an endnote. 7- Ref. [49]: correct typesetting of chemical elements in the title. 8- Ref. [54]: there is still an issue with the first name(s) of the author.

  • validity: high
  • significance: good
  • originality: high
  • clarity: good
  • formatting: excellent
  • grammar: excellent

Author:  Thomas Fischer  on 2022-08-06  [id 2713]

(in reply to Report 3 on 2022-08-04)
Category:
answer to question

The referee is correct that there is a two fold degeneracy in antiferromagnetic Neel states. The final state in our simulations consist of different states with odd sites magnetized downwards and even sites magnetized upwards adjacent to the opposite state and they are separated by domainwalls that cannot disappear like those between similar regions. We will mention this in a revised manuscript. We have also corrected the typos 1-8 in our corrected third version that we will resubmit as soon as being asked to do so by the editor. Thank you again for the helpfull review.

---

## Round 2 · Author Response

Dear editors of SciPost Physics,

Please find enclosed our revised manuscript "Adiabatic and irreversible classical discrete time crystals" by Adrian Ernst, Anna Rossi, and Thomas M. Fischer, for your consideration for
publication in SciPost Physics.

We are happy the reviewers like our manuscript. Both referees found parts of our manuscript difficult to follow. We have therefore significantly enlarged section 2 of the manuscript by giving more details that we hope will clarify better the topological arguments made in this section. We have followed all other recommendations and ammended the manuscript accordingly. In our response to the referees (referee remarks in italic, response in green and changes to the manuscript in blue) you will find a detailed point to point response to both referees. We hope that with these changes our manuscript is suitable for publication in SciPost Phys.

Sincerely Yours,

Thomas M. Fischer on behalf of all the authors.

---

## Round 2 · List of Changes

\textbf{Reviewer B: (Report)}

\textcolor{ao}{We thank the Referee for the constructive criticism and for the for the positive judgement. We respond to all points raised by the Referee in the following.}

\textit{The authors propose a realization of classical discrete time crystals in a system of paramagnetic colloidal particles that are placed above a magnetic hexagonal pattern and exposed to an external field that changes its direction periodically.}

\textit{ I have to admit that this work is a bit peripheral to my expertise. Thus, although it does seem interesting, I have confess that I have a bit of a hard time following it.}

\textcolor{ao}{We are happy the referee find our manuscript interesting.}

\textit{ Part of the issue may be related to the figures that are glossy, but nevertheless difficult to understand.}

\textit{ Figure 1 is particularly difficult to understand. If its description would mention specific parts (left, middle, or right panel), this would probably already help. Likewise, section 2 is quite abstract and thus not easy to read, at least not for a non-expert.}

\textcolor{blue}{We have marked Figure 1 with a) b) and c), significantly enlarged the figure caption and also enlarged section 2 with now referring to the different parts of the figure in more detail. The proof of an adiabatic topological time crystal is general and abstract. We hope together with the referrals to the example shown in figure 1 that we have now added to ease the reader to follow the arguments such that section 2 is now better to understand}

\textit{Another detail is recurrent terminology such as "north pole". I only found a short explanation on page 5 during a careful second reading. At the same time, the angles $\vartheta$ and $\varphi$ are not properly introduced. Conversely, line 2 of the caption of Fig. 5 mentions a direction that is "normal to the pattern", which might be identical to "north".}

\textcolor{blue}{We have included the definition of north (equivalent to normal to the pattern) , as well as the definition of the coordinates of control space $\vartheta_\cal C$ and $\varphi_\cal C$ in Figure 2, where we introduce our colloidal model system and we also introduce this terminology on page 5 of the revised manuscript}

\textit{ The caption of Fig. 7 should mention that this space-time picture is for a chain of honeycomb cells.}

\textcolor{blue}{Thank you, we have followed your advise}

\textit{ The five accompanying video clips are extremely useful to clarify and illustrate the findings of the present work. However, I believe that it would be good to emphasize that these videos are not just a supplementary piece, but actually an important part of this work. At the same time, the source in Ref. [49] needs to be specified more clearly. It may be Ok to use arXiv for them, but since this is not SciPost, this should be stated explicitly.}

\textcolor{blue}{We have set links to the arXiv in the manuscript, and the reference. We also emphasize that these videos are not just a supplementary piece, but actually an important part of this work in the text on page 9.}

\textit{ Content-wise, the middle panel of Fig. 1 does not factorize into a control space $\cal C$ and an action space} {${\cal A}$, as the notation ${\cal C}\otimes{\cal A}$ seems to suggest.}

\textcolor{ao}{The stationary manifold $\cal M$ in our example is a two dimensional manifold in a three dimensional curved space ${\cal C}\otimes{\cal A}$. We can visualize the stationary manifold only by deforming it in a way that preserves its topology such that it fits into Euclidian space. Such deformed version is shown in b). Naturally by only showing a topological equivalent version of the stationary manifold some properties such a visual factorization of the manifold into components in $\cal C$ and components in $\cal A$ get lost. The alternative is to embed the three dimensional manifold into a five dimensional Euclidian space and plot everything correctly. Unfortunately our eyes are not made for 5d-vision. This is why we have to defrom the manifold into a topological equivalent version that is embedded in 3d space.} \textcolor{blue}{We explain this in the figure caption to figure 1 and in the text.}

\textit{ A further comment on content is the emphasis on the behavior at the edge in section 5.2. After all, a spatial crystal should break translational symmetry spontaneously while an edge breaks it explicitly. I would be grateful if the authors could clarify this point.}

\textcolor{ao}{Yes the edge breaks both spatial translation symmetries of the bulk crystal explicitly. However the edge is a spatial edge and there is no edge in time. Therefore the edge does not explicitly break the time translational symmetry. This is broken via the response of the colloids in the edge cell. The crystal remains a time-crystal at the edge, i.e. a time periodic structure with broken time translational symmetry.}\textcolor{blue}{ We have added a sentence: "Note that at the edge of a Kagome lattice the translation symmetry normal to the edge is explicitly broken, while the translation symmetry in time parallel to the edge is only broken due to the time crystalline response of the colloids inside the lattice." on page 13, and remind the reader in the caption to Fig.6 that there is no edge in the direction of time.}

\textit{ Final remark on content: At the very end of the text, the authors say "All integrations were performed numerically". I think that the authors should give a few more details, and in particular specify the integration algorithm.}

\textcolor{blue}{We have included the integration algorithm (Euler algorithm) into the description in the appendix on page 17.}

\textit{ I would like to see the above points clarified before I commit on a definite recommendation for SciPost Physics.}

\textit{There are a few further minor issues that I list among the "Requested Changes".\ 1- Make the description of Fig. 1 more accessible.\ 2- Try to explain the content of section 2 better to a non-expert.\ 3- Define the angles $\vartheta$ and $\varphi$ properly and use this opportunity to specify the meaning of the "north pole".\ 4- Clarify if the "normal to" on line 2 of the caption of Fig. 5 is synonymous to "north".\ 5- The caption of Fig. 7 should mention that this space-time picture is for a chain of honeycomb cells.\ 6- Check if the terminology "nuclei" (several) versus "nucleus" (one) is used correctly in the video clip "adFig7.mp4".\ 7- Emphasize in the text that the "supplementary videos" are an integral part of this work.\ 8- Specify the source of in Ref. [49] (arXiv?) more clearly.\ 9- Clarify the notation ${\cal C}\otimes{ \cal A}$, or possibly rather the meaning of $\cal C$ and $\cal A$, as appropriate.\ 10- Clarify the relevance of the edge in section 5.2 (Fig. 6(b)): in which sense is the resulting state still a crystal?\ 11- Expand the comment "All integrations were performed numerically" at the end of the Appendix. In particular, specify the integration algorithm.\ 12- Page 11: eliminate abbreviations that are not used again ("DO TC") or only a second time ("FDO").\ 13- Page 11: Using angles $\langle , \rangle$ rather than less and bigger signs < > for expectation values would improve readability.\ 14- Still page 11: The "timely" should probably rather be a "time" or "time-like".\ 15- Caption of Fig. 6: I think that "spatial" would be preferred spelling over "spacial".\ 16- Format Eq. (A3) such that the curly bracket "{" stretches across both alternatives. 17- Last line of text: "effect" $\rightarrow$ "affect".\ 18- Properly upper-case names in the titles of Refs. [16,18,20,27,29,36,38,56].\ 19- Correct the spelling of the name of the author of Refs. [52,53]: "T. Prosen".\}

\textcolor{ao}{We have already explained our changes concerning points 1-5,7-11 in response to the report above }\textcolor{blue}{ and we followed the requests 6,12-19}

\textcolor{ao}{We appreciate the careful report that helped improve our manuscript}

\textbf{Reviewer A:}

\textit{The paper addresses an important aspect of topology in space and time to generate a nontrivial dynamic response of the system on external periodic driving. The paper introduces concepts and approaches using the example of a magnetic colloidal system driven above the patterned magnetic substrates to demonstrate a rich variety of topological time crystalline and non-time crystalline phases phenomena available. I was humbled by the complexity of the system. The paper is well written and suitable for publication in SciPost.}

\textcolor{ao}{We thank the refere for his positive report.}

\textit{ Requested changes}

\textit{the paper introduces a lot of variables that are needed for the story, but some of them are not defined in time. For instance, on page 8 when defining $\phi_\cal A$ authors use vector $\mathbf{a}_1$ that is not defined right away and only on page 10. }

\textcolor{blue}{We now define the unit vectors in Figure 2. and on page 5.}

\textcolor{ao}{We appreciate the careful report that helped improve our manuscript}

---

## Editorial Decision

published